# Compositional Hardness of Code in Large Language Models - A Probabilistic Perspective

## Abstract

A common practice in large language model (LLM) usage for complex analytical tasks such as code generation, is to sample a solution for the entire task within the model's context window. Previous works have shown that subtask decomposition within the model's context (chain of thought), is beneficial for solving such tasks. In this work, we point a limitation of LLMs' ability to perform several subtasks within the same context window – an in-context hardness of composition, pointing to an advantage for distributing a decomposed problem in a multi-agent system of LLMs. The hardness of composition is quantified by a generation complexity metric, *i.e.*, the number of LLM generations required to sample at least one correct solution. We find a gap between the generation complexity of solving a compositional problem within the same context relative to distributing it among multiple agents, that increases exponentially with the solution's length. We prove our results theoretically and demonstrate them empirically.

## 1 Introduction

Large language models (LLMs), based on the transformer archietecture (Vaswani et al., 2017), have become very efficient problem solvers in many domains, such as broad-scoped question answering, writing assistance, teaching, and more (Brown, 2020; Radford et al., 2019; OpenAI, 2023; Bubeck et al., 2023; Nori et al., 2023; West, 2023). Yet their analytical skills, such as coding capabilities, are slow to develop - Chen et al. (2021b); Li et al. (2022a); Alp (2023); Ridnik et al. (2024) show that even with millions of generations, LLMs may not produce a single correct solution to competitive coding problems. Zhuo et al. (2024), provide a benchmark for complex coding tasks, and show that SOTA LLMs are not yet capable of following complex instructions to use function calls precisely, with a performance significantly lower than human performance. Dziri et al. (2024) show that LLMs solve compositional tasks such as long multiplication and dynamic programming without developing systematic problem-solving skills.

One way to empower LLMs in analytical tasks, is to use subtask decomposition, otherwise known as chain of thought (COT) - a method in which an LLM breaks down a problem to smaller, more manageable tasks, solves them, and integrates it into a solution. The method has been empirically demonstrated by Wei et al. (2022), that show reasoning capabilities of language models improve when they are prompted to break down a task. Its efficiency has also been studied theoretically - Wies et al. (2022); Malach (2023) prove that through the autoregressive nature of language models, problems that cannot be solved directly, can be solved by subtask decomposition, Merrill & Sabharwal (2023) prove that while transformers are limited in the computational problems they can solve directly, using a polynomial number of intermediate steps, they can represent any polynomial time Turing Machine, and Sanford et al. (2024) provide a similar result on expressing more complex arithmetic circuits using more steps of COT.

Yet even with task decomposition, there is a limitation to transformer based models on analytical tasks with COT, due to their limited ability to compose functions - Sanford et al. (2024) show single layer attention can only learn pairwise relations between tokens, limiting the ability to integrate intermediate steps for a fixed model size, and the work of Peng et al. (2024) shows that iterative composition over a domain is limited by the size of the model, even when COT is used. Xu et al.

(2024), show limitations of compositionality on simple linguistic tasks. Thus while theoretically possible, some tasks require an arbitrarily long COT for an LLM to solve. However, in practice, LLMs are limited in their context length - beyond the constraint of context length during training, Hsieh et al. (2024) introduce the RULER benchmark, for measuring the utility of LLMs on long context tasks. They show that in practice many models can perform tasks only on a much shorter context length than they were trained on. Similarly, Liu et al. (2024) show LLMs cannot fully use all the information within their context and Ebrahimi et al. (2024) show empirically that LLMs are limited in random access to tokens within the context, in the bit parity problem. Consequently, even though COT can in theory allow an LLM to solve arbitrarily complex analytical problems, in practice, they will be limited by the effective context length.

A rising approach to remedy this limitation is to solve problems through the use of multi-agent systems, that tackle complex problems through the use of agents, where each agent is an LLM instance that solves a different aspect of the problem. While it has been used for simulating social interactions, (Park et al., 2023; Li et al., 2023; Pang et al., 2024), it has also been shown as an effective tool for analytical problem solving. This can be done by decomposing a large task and distributing the sub-tasks between agents. Ishibashi & Nishimura (2024) use this method for building large code bases and Liu et al. (2023), use a dynamic LLM-agent network for solving code problems and analytical tasks.

In this work, we theoretically study a compositional hardness of coding problems originating from context processing limitations of LLMs, and the resultant effectiveness of a multi-agent system over a single model instance in composite coding problems. We model a composite coding problem using a pair of simpler coding problems, such that the solution to the problem can be obtained from concatenating the solutions to the problem pair. Such solutions are realized in a chain of thought process, in which the model solves a complex problem by breaking it down to smaller subproblems. The model's usefulness on a coding task, is quantified by a generation complexity metric (definition 2) - the number of LLM generations required to sample at least one correct solution. The appeal of this metric, is that due to the existence of code testing units, it suffices to turn an LLM into a good program candidate generator and simply output a candidate that is correct (Kambhampati et al.; Thakur et al., 2023; Luo et al., 2024). The single LLM instance solves the entire problem, while the multi-agent system is comprised of two agents, each is an LLM instance tasked with solving one of the problems in the pair. Thus the LLM's helpfulness in the single instance case is the generation complexity for the composite problem, while in the multi-agent case, it is the product of generation complexities of the pairs of problems, as a correct solution is attained when independently sampling a correct solution to each problem.

We theoretically model an LLM as an autoregressive model, where a solution is sampled token by token, based on the hidden representations. When combining two problems whose solutions are grammatically similar but semantically different (such as different coding problems), we assume this combination injects noise into the model's representations during solution generation for each sub-problem (assumption 1), which we denote as screening. We show that a compositional problem may have an exponential gap in generation complexity relative to the product of sub problems' generation complexities (theorem 1), meaning an exponential hardness of composition in-context. Essentially, the model is less capable of solving two problems if they are presented within the same context, than if presented in separate contexts. This points to an advantage of decomposing a problem not only within the same LLM context, but to distribute the problems among multiple agents (*i.e.* solve each sub-task within a different context). Additionally, this result provides a view of the model's effective context length through the lens of screening, which is the model's ability to isolate the relevant context at each decoding step – additional irrelevant context may reduce model's performance on other tasks within the context window exponentially with length. We validate our assumptions and results experimentally on Llama 3 by constructing composite code problems.

## 2    RELATED WORKS

**LLMs as solution candidate generators in programming:**    Empirical works have shown that while contemporary LLMs struggle to solve complex programming tasks if one samples only a single solution per problem, they become much more useful if one samples multiple solutions and filters out correct ones with testing units (Chen et al., 2021b; Li et al., 2022a; Alp, 2023; Ridnik

et al., 2024). These works show that for some problems, out of thousands of generated solutions, only a handful are correct. Hence empowering an LLM with a large sampling budget drastically boosts its capabilities. As such, a relevant approach to quantify an LLM's usefulness in such tasks is the expected number of programs one needs to sample from it to obtain a correct solution. This motivates our definition of generation complexity (definition 2), and in this work, we primarily focus on providing theoretical results on the relative generation complexity in compositional coding tasks.

**Theoretical results on composition:** Previous works on multi-step function composition and sub-task decomposition in language models primarily focus on expressing and learning a deterministic function that solves well defined mathematical tasks, such as solving the bit parity problem (Wies et al., 2022), expressing and learning Turing machines (Merrill & Sabharwal, 2023; Malach, 2023), or composing a single mathematical function an arbitrary number of times (Peng et al., 2024). These do not necessarily translate in an interesting manner to code generation. In such results, for a given problem, the model provides a deterministic answer which is either correct or incorrect. This deviates from the practical use of LLMs for code generation mentioned above that is based on probabilistic sampling of multiple solutions and filtering the correct ones. Thus previous works cannot capture the usefulness of LLMs on code, as SOTA LLMs generally do not deterministically provide correct solutions to complex coding problems. In this work, we deal with probabilistic autoregressive generation of solutions to problems, which are more applicable to the practice of LLM code generation. We provide a softer more informative result for *how hard* it is to compose problems with the generation complexity metric, which is not possible through previous approaches.

**Effectiveness of LLMs in utilizng long context:** Previous empirical works (Hsieh et al., 2024; Liu et al., 2024) have shown LLM performance on tasks such as retrieval degrade when performed on longer contexts, and that models may be limited in their random access to tokens within the context (Ebrahimi et al., 2024). In this work, we study the degradation of language models on compositional coding tasks through the lens of noise that pieces of context from different subtasks insert into the generation process. Differently from above mentioned works, in our results the context does not necessarily have to be very long in order for the model performance to drop, but rather that grammatically similar but semantically different pieces of the context may "confuse" the model and harm code generation even in shorter contexts.

## 3 FRAMEWORK

### 3.1 GENERATION COMPLEXITY

We focus on coding problems, meaning each problem is written in natural language and is solved by a function, and the goal is for the model to generate a code that implements it.

**Definition 1** *Let $L$ be a programming language. Let $x$ be a natural language description of a problem, that is solved by function $f$, then a computer program $y$ is a correct solution to $x$, if it implements $f$.*

We formally define the generation complexity of a of a problem as the inverse success rate of a model to generate a correct solution to the problem:

**Definition 2** *For a problem $x$ and a natural language distribution $P$ over $V^*$, the generation complexity of $x$ w.r.t. $P$, is:*

$$N(P, x) = \frac{1}{\sum_{y \in correct\ solutions} P(y|x)} \tag{1}$$

*Where $V^*$ is the Kleene closure of the vocabulary $V$.*

Intuitively, the generation complexity is the number of program candidates needed to be sampled from the conditional distribution $P(\cdot|x)$ to get a program that solves the problem.

In this work, we consider simple compositional problems, which are decomposable to two smaller problems, and their concatenated solutions can be used to solve the compositional problem (but note that it can be expanded to any number of compositions). This captures cases such as a composition

of two code functions, or simple manipulations on outputs of code functions (*e.g.* concatenation, product, etc.), but more broadly, these types of solutions to compositional problems are realized in the use of chain of thought, where a model solves a complex problem in steps comprising of smaller subproblems. For a compositional problem $x$, decomposable to $x_1, x_2$, we quantify the in-context hardness of composition as the ratio of generation complexity to the full problem $N(P, x)$ with the product of generation complexities for the sub-problems $N(P, x_1) \cdot N(P, x_2)$. If the latter is much smaller, this points to an advantage for distributing the sub-problems between different instances of an LLM (multiple agents), as it would require fewer generations to sample a correct solution. This defines a compositional hardness of coding problems that originates from context processing limitations, as the compositional problem can be mathematically equivalent to solving both problems, yet seeing them both in the same context reduces the model's performance on them.

## 3.2 SCREENING IN AUTOREGRESSIVE MODELS

Here we introduce a source of hardness in code composition based on the autoregressive nature of LLMs. Typically, latent representations of the model contain information about the context beyond the next token prediction, *e.g.* the structure of the solution to problems (Ye et al., 2024). Thus when composing two code problems, we expect the representations during the generation of the second program to contain information about the first program and vice versa, which is grammatically similar (same programming language) but semantically very different. As a result, this additional information creates noise that can harm the generation process.

Formally, we denote by $r^{(L)}(x)$ the model's last hidden layer representation of the prompt $x$, by $U$, the model's unembedding matrix (hidden dimension to vocabulary). The logit of the $i$'th token is thus defined as the token's score before the softmax operation: $\langle r^{(L)}(x), U^T e_i \rangle$, where $e_i$ is the one-hot vector of the token. The probability distribution at each decoding step is the softmax applied to the logits, $P_{LLM}(i|x) = softmax(\langle r^{(L)}(x), U^T e_i \rangle)$.

In the process of generating a solution to a compositional code problem, $x$, that is implicitly or explicitly decomposable to $x_1$ and $x_2$, the model will implement a solution to the first part $y_1$ and then to the second part $y_2$. The sequence is generated based on the hidden representations. Informally, we expect the representation of the solution to the first problem, $y_1$, within the compositional problem, $x$, to be a noisy version of the solution's representation in the non-compositional problem, $x_1$:

$$r^{(L)}(x \oplus y_1) = r^{(L)}(x_1 \oplus y_1) + noise \tag{2}$$

Similarly, the representation of the second problem's solution, $y_2$, in the compositional problem, $x$, is expected to be a noisy version of the solution's representation in the non-compositional problem, $x_2$:

$$r^{(L)}(x \oplus y_1 \oplus y_2) = r^{(L)}(x_2 \oplus y_2) + noise \tag{3}$$

Essentially, this means the model attempts to generate the same solutions as in the non-compositional case, but noise may interfere in the process. The projection of this noise onto the dictionary creates noise in the logits during decoding, which can lead the model to make mistakes. It is worth mentioning that while theoretically after generating $y_1$, it may serve as an in-context example for $y_2$, in practice when composing two complex problems that are semantically different (*e.g.* dynamic programming vs randomized algorithms), there is no reason for one to enhance the success rate of the other.

As the two problems $x_1, x_2$ and their solutions $y_1, y_2$ may be different semantically, we do not expect the noise to "push" the model towards the correct solutions more than to incorrect solutions, thus when projected onto the vocabulary, $V$, the noise on the logit of the correct token minus the noise on an incorrect token, $\langle noise, U^T e_{i_{correct\ next\ token}} \rangle - \langle noise, U^T e_{i_{an\ incorrect\ next\ token}} \rangle$, should be symmetric on average. Additionally it should be bounded within some range $[-M, +M]$, as it changes the hidden representation to a finite extent. In practice, we only expect this to be true for the high probability tokens, as the vocabulary $V$ is very large, and some low probability tokens may be systematically enhanced. To avoid this issue, we make our assumptions only on the weighted average of the noise, where the weights are given by the probability mass that the model assigns them. This way, low probability tokens with asymmetric noise or large norms receive low weight and are averaged with the noise of other tokens. Denote by $P(i|context)$ the probability assigned to the $i$'th token given the context. We will make our assumptions on the *weighted average of the noise on the incorrect*

*token logits minus correct token logits*:

$$X = \frac{\sum_{i \in V \setminus \{correct\ next\ token\}} P(i|context)\langle noise, U^T e_i - U^T e_{i_{correct\ next\ token}}\rangle}{\sum_{i \in V \setminus \{correct\ next\ token\}} P(i|context)} \tag{4}$$

**Assumption 1** *Denote by $X$ the weighted noise on the logits as defined in equation 4. We assume that at every given decoding step it is a continuous, symmetric random variable and bounded within $[-M, +M]$ for some $M > 0$.*

In experiment subsection 5.3 we show the noise satisfies these assumptions, with $M \approx 3 - 4$.

### 3.3 EFFECT OF NOISE ON DECODING

While the noise onto the logits, $X$, averages to zero, its effect on the decoding process does not. The probability of each token in a decoding step is changed to:

$$P(i|context) \rightarrow P'(i|context) \leq \frac{P(i|context)}{P(i|context) + (1 - P(i|context))e^X} \tag{5}$$

The denominator, $P(i|context) + (1 - P(i|context))e^X$ can be thought of as a renormalizing term, which redistributes the probability of the tokens. For $X = 0$, the token's probability does not change, for $X < 0$ it increases and for $X > 0$ it decreases. See appendix D for derivation and intuition. Note that if $P(i|context) \in (\epsilon, 1 - \epsilon)$ for $\epsilon > 0$ (the model has finite confidence), then *on average, the noise decreases the probability of a correct continuation $P(i|context)$*, by a factor of $\exp(-\Delta(\epsilon, X))$, where $\Delta$ is the renormalizing term's mean:

$$\Delta(\epsilon, X) := \mathbb{E}_X[\log(\epsilon + (1 - \epsilon)e^X)] \tag{6}$$

Intuitively $\Delta$ is the average renormalization of the correct token's probability. In experiment subsection 5.3, we calculate $\Delta(\epsilon, X)$ empirically as a function of $\epsilon$, and find that for $\epsilon = 0.1$ for example, $\Delta \approx 0.2$. The consequence of this, is that on average, most long sequences have their probability reduced by the noise, while few random long sequences have their probability greatly enlarged. Since for long coding problems most sequences are incorrect, the probability of a correct solution getting enlarged is small. We formally show this in the next section. To do so, we will use concentration inequalities, for which we note that:

$$|\log(\epsilon + (1 - \epsilon)e^X)| < M \tag{7}$$

Thus the renormalizing term's variance is also bounded:

$$\sigma^2(\epsilon, X) := Var_X[\log(\epsilon + (1 - \epsilon)e^X)] \leq M^2 \tag{8}$$

## 4 RESULTS

Here we show that composing two coding problems can be significantly harder than solving each on its own. A natural quantification of compositional hardness is the gap between generation complexity of the problem's components and the complete problem. For example, we say that composition is hard if:

$$N(P, x) \gg N(P, x_1) \cdot N(P, x_2) \tag{9}$$

While it is easy if:

$$N(P, x) \approx N(P, x_1) \cdot N(P, x_2) \tag{10}$$

The rational behind this, is that $N(P, x_1) \cdot N(P, x_2)$ is the number of attempts required to independently sample a correct solution to $x_1$ and to $x_2$, while $N(P, x)$ is the number of attempts required to sample a solution to problem that integrates both problems. In an easy composition scenario, the model solves the sub-problems to the best of its abilities as it would if each was solved independently. In a hard composition scenario, seeing both problems combined reduces its performance on each sub-problem, and it is better to use a multi-agent system, by feeding the model the subproblems in different contexts and sampling solutions independently.

As there are typically more incorrect solutions to coding problems than correct ones, the random noise inserted into the logits generally harms the model's performance. The following lemma quantifies an exponential decrease in the model's probability of a correct solution to a compositional problem, $P(y_1 \oplus y_2|x)$, relative to the probabilties of the sub-problem solutions $P(y_1|x_1) \cdot P(y_2|x_2)$:

**Lemma 1** *Let $\epsilon, \delta \in (0, 1)$, and $M > 0$. Let $x$ be a compositional problem and $y_1 \oplus y_2$ a solution, with $x_1$, $x_2$ being the corresponding sub-problems. Suppose that the noise injected to the logits as defined in equation 4, satisfies assumption 1, and that the probability assigned to the correct token at each decoding step is bounded within $[\epsilon, 1 - \epsilon]$. Then there exist strictly positive noise dependent constants $\Delta$ (as defined in equation 6) and $c(\Delta, M, \sigma)$ (with $M$ and $\sigma$ as defined in equations 7 and 8), such that if the solution length satisfies $|y_1| + |y_2| > c \ln \frac{1}{\delta}$ we have with probability of at least $1 - \delta$ that:*

$$P(y_1 \oplus y_2 | x) \leq P(y_1 | x_1) \cdot P(y_2 | x_2) e^{-\frac{\Delta \cdot (|y_1| + |y_2|)}{4}} \tag{11}$$

*Where $P(y_1 \oplus y_2 | x)$ is the probability of producing the answer $y_1 \oplus y_2$, given context $x$. The constant $c = \frac{M^2}{\sigma^2 \cdot h(\frac{3\Delta \cdot M}{4\sigma^2})}$, with $h(x) = (x + 1) \log(x + 1) - x$.*

The proof is presented in appendix A. The intuition behind this result is that as there are typically more incorrect choices to make when generating code, random noise usually reduces the probability for sequences with finite confidence. Thus most sequences get their probability reduced, while very few random sequences get a large increase, and these are usually not correct solutions.

The assumption on bounded probability for the correct token $[\epsilon, 1 - \epsilon]$ implies we are considering solutions where the model has high but limited confidence in each decoding step. While LLMs are often very confident in "obvious" next steps (line break, etc.), in practice, during generation, nucleus sampling is commonly used, where sampling only occurs if the model is not overly confident. *e.g.* with $p = 0.95$, if a token's probability $P$, is larger than $0.95$, then the probability is rounded 1. Thus when considering probabilities of sequences, it suffices to look only at decoding steps where the model is not too confident, hence we can consider $\epsilon \approx 0.05$.

In subsection 5.3, we see empirically that for $\epsilon = 0.1, \Delta \approx 0.2, \sigma \approx 1.5, M \approx 4$, making $c \approx 200$, thus for solutions with length $> 200$ tokens, the results apply.

In practice, there may be multiple solutions to the same problem, *e.g.* multiple implementations of the same function. So it is necessary to take all of them into account when considering the generation complexity. With this taken into account using a union bound, we obtain the following result of an exponential gap in generation complexity between a composition of problems and the sub-problems, indicating a compositional hardness that is exponential in the solution's length:

**Theorem 1** *Let $\epsilon, \delta \in (0, 1)$, and $N, M > 0$. Let $x$ be a compositional problem, with $x_1$, $x_2$ being the corresponding sub-problems. Denote by $L_1, L_2$ the minimal solution length to $x_1, x_2$ respectively, and the total number of solutions to $x$ by $N$. Under the assumptions of lemma 1, there exist strictly positive noise dependent constants $\Delta$ (as defined in equation 6) and $c(\Delta, M, \sigma)$ (with $M$ and $\sigma$ as defined in equations 7 and 8), such that if the minimal solution length $L_1 + L_2$, satisfies $L_1 + L_2 > c \ln \frac{N}{\delta}$, then with probability of at least $1 - \delta$ the generation complexity (definition 2) satisfies:*

$$N(P, x) \geq N(P, x_1) N(P, x_2) \cdot e^{\frac{\Delta \cdot (L_1 + L_2)}{4}} \tag{12}$$

The proof is presented in appendix B. We see that longer problems become harder to solve due to the noise injected into the decoding steps by previously generated tokens. This shows that a model that fully utilizes its context in decoding (*i.e.* the next token probability distribution is explicitly a function of all the previous tokens), can have a hard time mixing different concepts due to the screening effect. This result implies that for long coding problems, it is more beneficial to distribute sub-tasks between different instances of the LLM, and not expose it to the full context. Additionally, this result provides a view of the model's effective context length through the lens of screening – additional irrelevant context may reduce model's performance on other tasks within the context window exponentially with length. The intuition to the result is that there are more incorrect solutions than correct ones, so the random noise usually reduces their total probability mass.

We note that for the union bound of theorem 1 to hold, we need the number of solutions to be bounded by an exponential in the length of the solution, $L_1 + L_2$:

$$N < \delta \cdot \exp\left((L_1 + L_2) \cdot \frac{\sigma^2}{M^2} h\left(\frac{3\Delta \cdot M}{4\sigma^2}\right)\right) \tag{13}$$

Typically, we expect the number of solutions to a problem to grow exponentially with the length of the shortest solution, $y$:

$$N \sim \exp(c \cdot |y|) \tag{14}$$

As each line of code can be implemented slightly differently, lines may be interchanged, different variable names, etc. Still, the exponential's coefficient is expected to be small as most sequences are not correct solutions to the problem due to the constraints between the tokens (*e.g.*, once a variable name is chosen, it is fixed throughout the solution). With the empirical values calculated in subsection 5.3, we see empirically that the coefficient of the exponential number of solutions is $\approx 0.005$, thus for a solution of length $L_1 + L_2 = 1,600$ tokens, we have $N < \delta \cdot \exp(8) = 3,000 \cdot \delta$ solutions.

## 5 EXPERIMENTS

In this subsection we test the assumptions and results of our theoretical part. We create simple compositional coding problems with pairs of problems from the Human Eval benchmark (Chen et al., 2021a) and code contests dataset (Li et al., 2022b). First, we show the results of theorem 1, stating that composition is typically harder than independently solving the subproblems. Then, we show explicit indications for the exponential length dependence of compositional hardness, by comparing probabilities of solutions with/without composition, as theoretically suggested in lemma 1. Finally, we look at our assumption 1 on the noise inserted into the logits of the second problem, and observe that it is indeed large enough to interfere with the decoding process (assumption 1). The experiments were performed on Llama-3-8B-Instruct (Dubey et al., 2024). In appendix F, we present results for Llama-3-70B-Instruct, to test dependence on model size. For additional experimental details, see appendix E.

### 5.1 GENERATION COMPLEXITY RESULTS

Here we test the actual generation complexity of an LLM to different problems corresponding to scenarios from our theoretical results. As proposed in the theoretical section, an LLM may be able to solve two problems with low generation complexity, but their composition might take a significantly larger generation complexity if the model was not explicitly trained on such a task. To test this, we built a set of composite problems based on the Human Eval benchmark (Chen et al., 2021a) and code contests dataset (Li et al., 2022b).

To create composite problems, we took pairs of problems from Human Eval and code contests, and created from each a problem whose solution requires to explicitly solve both problems. Additionally, for harder problems, we created compositions of problems from code contests dataset. We used the following two main templates (and an additional template for human eval presented in appendix E):

- Human Eval – Problem 1 and Problem 2 have an integer output. Composition is to solve both and print the product of their outputs.

> Complete the two following functions in python. Function 1: [Function 1 description]. Function 2: [Function 2 description]. Print the product of their outputs: Function 1 (input 1)* Function 2 (input 2).

- Code Contests – Problem 1 and problem 2 are problems from the dataset. Composition is to read the inputs sequentially and print the outputs sequentially.

> Solve the following two problems in python. Problem 1: [Problem 1]. Problem 2: [Problem 2]. The solution should read the inputs to the problems sequentially and print the outputs sequentially.

We denote the composite problem of $x_1$ and $x_2$ as $x_1 \oplus x_2$. While these compositions may appear artificial, their purpose is to demonstrate the "mental load" on the model during composition, which leads to the result of theorem 1. This serves as a lower bound for compositional hardness, due

to explicit concatenation of problems being the easiest form of composition, while more complex compositions can lead to worse performance. For each problem we sampled 200 solutions and evaluated the generation complexity as the inverse of percentage of correct solutions.

Figure 1 shows the cumulative distribution function (CDF) of the compositional hardness - the value along the $y$ axis is the CDF of $\frac{N(P,x)}{N(P,x_1)N(P,x_2)}$, *i.e.* percentage of compositional problems, in which $\frac{N(P,x)}{N(P,x_1)N(P,x_2)}$ is smaller than a given value. For example, with $a = 5$, the corresponding value on the $y$ axis, is the percentage of problems in which composition requires up to $\times 5$ more generations relative to solving independently. As can be seen, for most of the problems, composition generation complexity is larger than the product of generation complexities of the components (up to factors of $10 - 20$). As seen in theorem 1, we have:

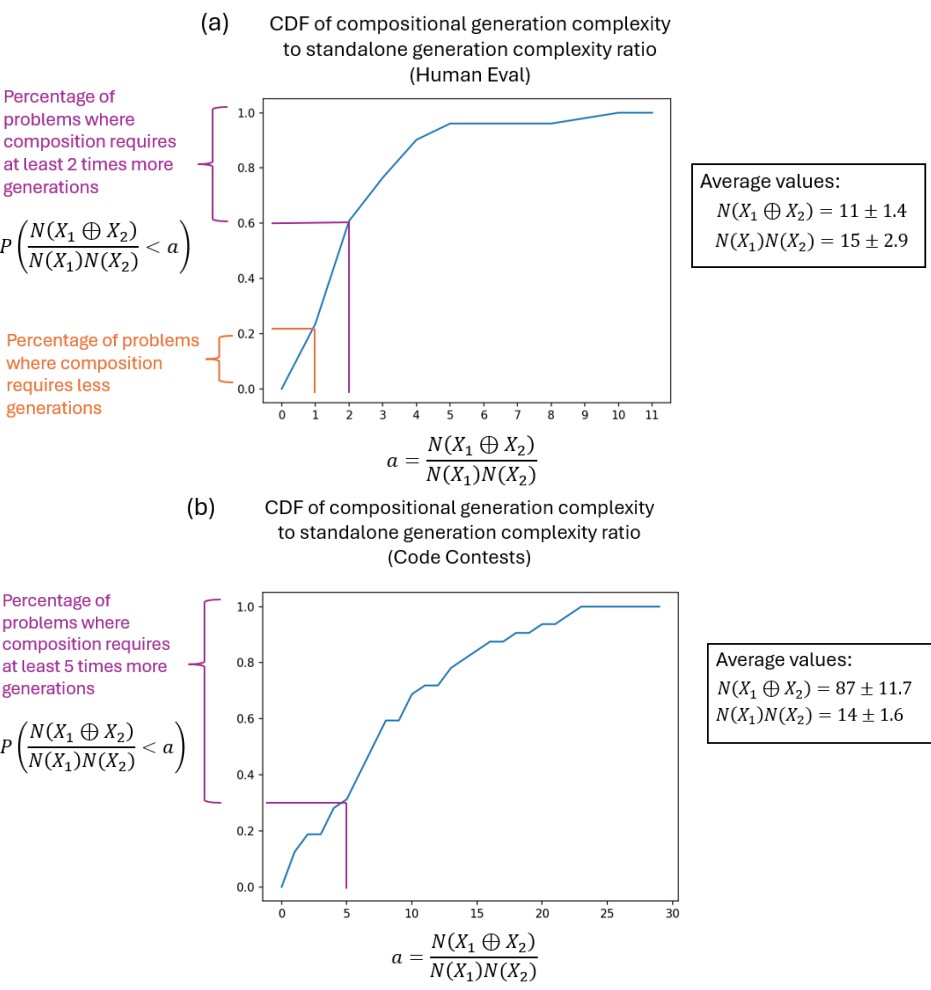

Figure 1: Cumulative distribution function for the ratio of generation complexity using composition, $N(P,x)$, to product of generation complexities for the standalone problems, $N(P,x_1) \cdot N(P,x_2)$ (corresponding to the multi-agent generation complexity). The x axis denotes values for the ratio of generation numbers required to solve the problem in the two cases (composition vs multi-agent), the y axis is the percentage of problems in which the ratio is no larger than this value (*e.g.* for $a = 5$, the y axis value is the percentage of problems where composition requires up to $\times 5$ more samples than the multi-agent case). (a) For the human eval composition. As can be seen in most of the cases, composition requires twice more samples, and for some problems 10 times more samples. (b) For the code contests composition. As can be seen the majority of problems have a factor of at least 5, and some up to 20.

$$N(P, x_1 \oplus x_2) \gg N(P, x_1)N(P, x_2) \tag{15}$$

Furthermore, we see that in code contests, where the problems typically have longer solutions (few hundreds of tokens), and the problems are harder thus the model is less certain, the compositional hardness is much larger. Assuming one only has access to an end-to-end verifier of the composite problem, it is more advantageous to generate independently for each problem then to generate for the composite problem, giving the advantage to a multi-agent system over a single LLM instance.

In appendix F, we present results for composition on Llama-3-70B-Instruct, and find that on the same code contests compositions, the compositional generation complexity improves significantly relative to its 8B counterpart (similar to 8B on Human Eval seen in figure 1a). However, with slightly harder compositions of code contest problems (that are still effectively concatenations of two problems), the 70B model's performance on composition drops significantly, with most compositions requiring over 5-10 times more generations than the non-compositional case. This hints larger models are more efficient at composition yet still suffer from this effect as compositional difficulty rises.

## 5.2 Exponential Length Dependence of Compositional Hardness

Here, we used the same compositions as in the previous subsection, and measured the difference in probabilities of correct solutions with vs without composition as a function of the number of tokens in the solution. The "correct" solutions were taken as the canonical solutions of the dataset, as they are "neutral" in the sense that solutions generated by the model with/without composition may have different styles, and measuring the probability of these generated sequences may create an artificial bias in favor of one of the two.

As suggested by lemma 1, we expect to see on average:

$$\log \frac{P(y_1|x_1)P(y_2|x_2)}{P(y_1 \oplus y_2|x)} \geq \frac{\Delta}{4} \cdot (|y_1| + |y_2|) \tag{16}$$

Meaning that the log ratio of correct solutions without/with composition increases linearly with length. As can be seen in figure 2, an exponential increase in probability of the correct solution is observed without composition relative to with composition, as a function of length of the code. As the y axis is plotted in the log domain, the linear curve approximates $\Delta/4$, which takes a value of $\approx 0.05$. This means a decrease of $e^{-1}$ in success rate for every 20 tokens. This matches the typical increase in generation complexity seen in figure 1. In practice, $\Delta$ is likely larger for some compositions, as the average is over many sequences, yet still has high fluctuations.

While measuring probability of sequences not generated by the model is less reliable than the above experiment in 5.1, which shows the gap of compositional hardness in a real sampling scenario, this approach is useful to qualitatively show how in certain compositional scenarios, the probability of sequences are exponentially lower than non-compositional scenarios, as a function of code length, which is expected to hold during generation.

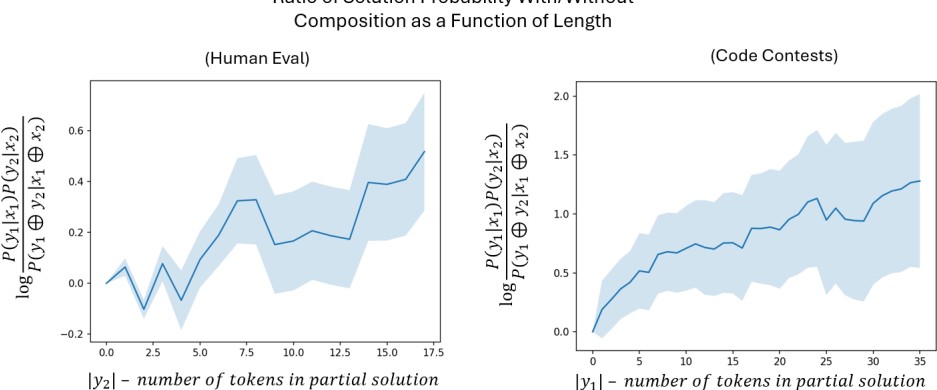

Figure 2: Ratio of correct solution probability with vs without composition. An exponential trend is observed as a function of length.

## 5.3 EXPERIMENTS ON ASSUMPTIONS

Here we look at our assumption of noise inserted into the logits (assumption 1).

**Noise distribution:** Using the same compositions as before, we measure the change to the logits of correct tokens with vs without composition. As in the theoretical assumption, we subtract this with the mean change in the logits of the incorrect tokens:

$$X = \frac{\sum_{i \in incorrect} P_i \langle noise, U^T e_i - U^T e_{correct} \rangle}{\sum_{i \in incorrect} P_i} \quad (17)$$

We calculate this over different sequences. As can be seen in figure 3, the change in logits creates a noise that is symmetric, bounded, $X < M \approx 4$, and has finite absolute deviation $E[|X|] > 0$, in accordance with assumption 1.

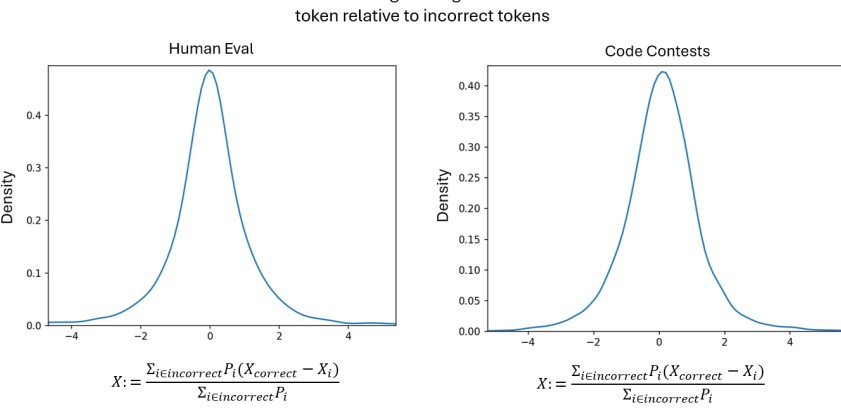

Figure 3: Change in logits of correct tokens minus incorrect tokens due to composition.

**Estimation of $\Delta(\epsilon, X)$ and $\sigma(\epsilon, X)$:** Here the goal is to provide a ballpark estimation to the theoretical constants. To estimate $\Delta(\epsilon, X)$ we approximate the noise as a Gaussian with mean 0 and try two values standard deviation $\sigma = 1$ and $\sigma = 2$, then calculate $\Delta(\epsilon, X) \approx mean_{\{X_i\}}[\epsilon + (1 - \epsilon)e^X]$. The values are presented in appendix G, and match the estimation of $\Delta$ in the range of $0.05$ to $0.2$ from the above subsection. Similarly, we estimate $\sigma(\epsilon, X)$, with typical values of $\sigma \approx 1 - 2$

## 6 DISCUSSION

In this work, we point a limitation of LLMs' ability to perform several sub-tasks within the same context window – an in-context hardness of composition, pointing to an advantage for distributing a decomposed problem in a multi-agent system of LLMs over using a single LLM instance. The hardness of composition is quantified by the generation complexity metric, *i.e.*, the number of LLM generations required to sample at least one correct solution. We found an exponential gap between generation complexity of solving a composition problem within the same context relative to distributing it among multiple agents. This is attributed to the transformer's nature of using all tokens in the context simultaneously for decoding, which inserts noise into generated sequences, caused from mixing in sub-tasks' latent representations (screening). Consequently, even if a model has the ability to perform two tasks, it may not be able to perform them both within the same context, in agreement with empirical work of Zhuo et al. (2024), showing SOTA LLMs still perform poorly on complex coding tasks. This provides a view of the model's effective context length through the lens of screening, which is the model's ability to isolate the relevant context at each decoding step – additional irrelevant context may reduce model's performance on other tasks within the context window exponentially with length. Lastly, we point that while advantegous, the use of multi-agent systems may introduce new challenges, such as coherence between agents, which we leave for future work.

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

# A PROOF OF LEMMA 1

Here we prove lemma 1. We formulate the detailed lemma:

**Lemma 2** *Let $\epsilon, \delta \in (0,1)$, and $M > 0$. Let $x$ be a compositional problem and $y_1 \oplus y_2$ a solution, with $x_1$, $x_2$ being the corresponding sub-problems. Suppose that the noise injected to the logits as defined in equation 4, satisfies assumption 1 - for all decoding steps, it is continuous, symmetric and bounded within $[-M, +M]$. Suppose further that the probability assigned to the correct token at each decoding step is bounded within $[\epsilon, 1 - \epsilon]$. Denote by $\Delta := \mathbb{E}_X[\log(\epsilon + (1 - \epsilon)e^X)]$ and $\sigma^2 := Var_X[\log(\epsilon + (1 - \epsilon)e^X)]$ the renormalizing term's mean and variance (as defined in equations 6 and 8 respectively). Under the assumption, $\Delta, \sigma$ are strictly positive, and if $|y_1| + |y_2| > \frac{M^2}{\sigma^2 \cdot h(\frac{3\Delta \cdot M}{4\sigma^2})} \ln \frac{1}{\delta}$, where $h(x) = (x+1)\ln(1+x) - x > 0$, we have with probability of at least $1 - \delta$ that:*

$$P(y_1 \oplus y_2|x) \leq P(y_1|x_1) \cdot P(y_2|x_2)e^{-\frac{\Delta \cdot (|y_1| + |y_2|)}{4}} \tag{18}$$

*Where $P(y|x)$ is the probability assigned to $y$ by the model, given context $x$.*

Thus $f$ from the main text is $f(\sigma, M, \Delta) = \frac{M^2}{\sigma^2 \cdot h(\frac{3\Delta M}{4\sigma^2})}$.

*Proof:*

We start by proving the following lemma:

**Lemma 3** *Let $p \in (0,1)$ and $X$ a random variable over real numbers. Denote by: $\Delta(p, X) = E\left[\log\left(p + (1-p)e^X\right)\right]$. Claim: For a continuous symmetric random variable, $X$, we have $\Delta(p, X) > 0$. Furthermore, if $p \in (\epsilon, 1 - \epsilon)$, then $\Delta(p, X) > \Delta(\epsilon, X) = \Delta(1 - \epsilon, X)$*

*Proof:*

Denote by $\rho(X)$ the density function of $X$, then:

$$E\left[\log\left(p + (1-p)e^X\right)\right] = \int_{-\infty}^{0} \log\left(p + (1-p)e^x\right)\rho(x) + \int_{0}^{\infty} \log\left(p + (1-p)e^x\right)\rho(x) = \tag{19}$$

Switching sign of the integration variable in the second term:

$$= \int_{0}^{\infty} \log\left(p + (1-p)e^{-x}\right)\rho(-x) + \int_{0}^{\infty} \log\left(p + (1-p)e^x\right)\rho(x) = \tag{20}$$

Using the symmetry condition on $X$:

$$= \int_{0}^{\infty} \log\left(p + (1-p)e^{-x}\right)\rho(x) + \int_{0}^{\infty} \log\left(p + (1-p)e^x\right)\rho(x) = \tag{21}$$

$$= \int_{0}^{\infty} \left(\log\left(p + (1-p)e^{-x}\right) + \log\left(p + (1-p)e^x\right)\right)\rho(x) = \tag{22}$$

Again, applying the symmetry of $X$:

$$= \frac{1}{2} \int_{-\infty}^{\infty} \left(\log\left(p + (1-p)e^{-x}\right) + \log\left(p + (1-p)e^x\right)\right)\rho(x) \tag{23}$$

The integrand is positive:

$$\log(p + (1-p)e^x) + \log(p + (1-p)e^{-x}) = \tag{24}$$

$$= \log\left(p^2 + (1-p)^2 + p(1-p)\left(e^x + e^{-x}\right)\right) = \tag{25}$$

$$\log\left(1 + p(1-p)\left(e^x + e^{-x} - 2\right)\right) \tag{26}$$

For $x = 0$, the argument inside the log equals 1, otherwise, the argument inside the log is always larger than 1, since $e^x + e^{-x} - 2 > 0$. Hence the integrand is positive everywhere except in $x = 0$, meaning the integral is positive for any continuous symmetric $X$. Thus, for any $p$ the expectation value is positive.

$$E\left[\log\left(p + (1-p)e^X\right)\right] = \Delta(p, X) > 0 \tag{27}$$

Furthermore it is symmetric around $p = 1/2$, and monotonic for $p \to 0$ and $p \to 1$:

$$\log(p + (1-p)e^x) + \log(p + (1-p)e^{-x}) = \tag{28}$$

$$= \log\left(p^2 + (1-p)^2 + p(1-p)\left(e^x + e^{-x}\right)\right) = \tag{29}$$

$$\log\left(1 + p(1-p)\left(e^x + e^{-x} - 2\right)\right) \tag{30}$$

As can be seen, it is monotonic with $p(1-p)$, and if $p \in (\epsilon, 1-\epsilon)$, it is minimal for $p = \epsilon, 1-\epsilon$.

**Proof of Main Lemma:**  We now move to the proof of the main lemma. We show the exponential growth with the length of the solution to the second problem, $y_2$ (the proof for the exponential dependence on $y_1$ is identical). Let us look at the probability that the model assigns the $n$'th token in the sequence $y_2$ in the compositional problem. It is given by the softmax on the projections of the final hidden layer representation on the vocabulary $V$ given the context:

$$P(y_2[n] | x \oplus y_1 \oplus y_2[:n]) = \frac{e^{\left\langle r^{(L)}(x \oplus y_1 \oplus y_2[:n]), U^T e_{y_2[n]}\right\rangle}}{e^{\left\langle r^{(L)}(x \oplus y_1 \oplus y_2[:n]), U^T e_{y_2[n]}\right\rangle} + \Sigma_{i \in [V] \setminus \{y_2[n]\}} e^{\left\langle r^{(L)}(x \oplus y_1 \oplus y_2[:n]), U^T e_i\right\rangle}} \tag{31}$$

According to assumption 1, $\left\langle r^{(L)}(x \oplus y_1 \oplus y_2[:n]), U^T e_i\right\rangle = \left\langle r^{(L)}(x_2 \oplus y_2[:n]), U^T e_i\right\rangle + X_i$, meaning the model is receiving a noisy version of the representation to the problem it is trying to solve, where $X_i$ is the noise onto the $i$'th token.

$$= \frac{P(y_2[n] | x_2 \oplus y_2[:n]) e^{X_{y_2[n]}}}{P(y_2[n] | x_2 \oplus y_2[:n]) e^{X_{y_2[n]}} + \Sigma_{i \in [V] \setminus \{y_2[n]\}\}} P(i | x_2 \oplus y_2[:n]) e^{X_i}} \tag{32}$$

For brevity, denote $P_0 = P(y_2[n] | x_2 \oplus y_2[:n])$, and $P_i = P(i | x_2 \oplus y_2[:n])$.

$$= \frac{P_0 e^{X_0}}{P_0 e^{X_0} + \Sigma_{i \in [V] \setminus \{0\}} P_i e^{X_i}} \tag{33}$$

Now, using the Jensen's inequality in the denominator:

$$\leq \frac{P_0 e^{X_0}}{P_0 e^{X_0} + (1 - P_0) e^{\Sigma_{i \in [V] \setminus \{0\}} \frac{P_i X_i}{1 - P_0}}} = \frac{P_0}{P_0 + (1 - P_0) e^{\Sigma_{i \in [V] \setminus \{0\}} \frac{P_i X_i}{1 - P_0} - X_0}} \tag{34}$$

Now, denote $X = \Sigma_{i \in [V] \setminus \{0\}} \frac{P_i(X_i - X_0)}{1 - P_0}$, according to assumption 1, it is a symmetric, continuous random variable, bounded between $[-M, +M]$. Rewriting the above as:

$$= P_0 e^{-\log(P_0 + (1 - P_0)e^X)} \tag{35}$$

So the correct token probability with composition $P_0'$ is decreased by the factor in the exponent $\to P_0 e^{\log(P_0 + (1-P_0)e^X)}$, relative to the probability without composition, $P_0$. For a full sequence, we apply the probability chain rule and obtain the following:

$$P(y_2 | x \oplus y_1) = P(y_2 | x_2) e^{-\Sigma_{i=1}^{|y_2|} \log\left(P_0^i + \left(1 - P_0^i\right)e^{X_i}\right)} \tag{36}$$

Where $P_0^i$ is the probability of the correct token in the $i$'th step without composition. Now, because $E_{X_i}\left[\log\left(P_0^i + \left(1 - P_0^i\right)e^{X_i}\right)\right] = \Delta\left(P_0^i, X_i\right) > 0$, from the above lemma, we get a sum of random variables with mean that is larger than zero. We will use a concentration inequality to bound it.

We start with bounding the random variables:

$$\log\left(P_0^i + \left(1 - P_0^i\right)e^{X_i}\right) < \log\left(P_0^i + \left(1 - P_0^i\right)e^{M}\right) \leq M \tag{37}$$

$$\log\left(P_0^i + \left(1 - P_0^i\right)e^{X_i}\right) > \log\left(P_0^i + \left(1 - P_0^i\right)e^{-M}\right) \geq -M \tag{38}$$

Also notice that since $P_0 \in (\epsilon, 1 - \epsilon)$, the above lemma implies:

$$\Delta\left(P_0^i, X_i\right) > \Delta\left(\epsilon, X_i\right) \tag{39}$$

Thus from linearity of the expectation value of the sum $S = \Sigma_{i=1}^{|y_2|}\log\left(P_0 + (1 - P_0)e^{X_i}\right)$ is:

$$E[S] = E\left[\Sigma_{i=1}^{|y_2|}\log\left(P_0 + (1 - P_0)e^{X_i}\right)\right] > |y_2|\cdot\Delta(\epsilon, X) \tag{40}$$

Similarly, if we denote $\sigma^2(\epsilon, X) := Var_X[\log(\epsilon + (1 - \epsilon)e^{X})]]$ (which is no larger than $M$), we can upper bound the variance of the sum:

$$Var[S] = Var\left[\Sigma_{i=1}^{|y_2|}\log\left(P_0 + (1 - P_0)e^{X_i}\right)\right] < |y_2|\cdot\sigma(\epsilon, X)^2 \tag{41}$$

This is because $Var_X[\log(p + (1 - p)e^{X})]] \leq Var_X[\log(\epsilon + (1 - \epsilon)e^{X})]]$ (see proof in appendix C)

We can then apply Bennet's inequality:

$$P\left(S - \mathbb{E}[S] < -t\right) \leq \exp\left(-\frac{Var[S]}{M^2}h\left(\frac{tM}{Var[S]}\right)\right) \tag{42}$$

Where $h(x) = (1 + x)\log(1 + x) - x$ and $M$ is the bound for each summand in $S$. Next, we take $t = \mathbb{E}[S] - \frac{\Delta(\epsilon, X)}{4}|y_2|$. Plugging this in the above yields:

$$P\left(S < \frac{\Delta(\epsilon, X)}{4}|y_2|\right) \leq \exp\left(-\frac{Var[S]}{M^2}h\left(\frac{tM}{Var[S]}\right)\right) \tag{43}$$

We note that $t = \mathbb{E}[S] - \frac{\Delta(\epsilon, X)}{4}|y_2| > \frac{3\Delta(\epsilon, X)}{4}|y_2|$ (from equation 40). Thus due to the (increasing) monotonicity of $h(\frac{tM}{Var[S]})$ w.r.t. $t$ (hence decreasing monotonicity in the exponent), we have:

$$P\left(S < \frac{\Delta(\epsilon, X)}{4}|y_2|\right) \leq \exp\left(-\frac{Var[S]}{M^2}h\left(\frac{3\cdot\Delta(\epsilon, X)|y_2|M}{4\cdot Var[S]}\right)\right) \tag{44}$$

Due to the (increasing) monotonicity of the exponent's argument in $Var[S]$ which is upper bounded by $|y_2|\sigma^2$, we get:

$$P\left(S < \frac{\Delta(\epsilon, X)}{4}|y_2|\right) \leq \exp\left(-\frac{|y_2|\sigma^2}{M^2}h\left(\frac{3\cdot\Delta(\epsilon, X)M}{4\cdot\sigma^2}\right)\right) \tag{45}$$

Looking at the complementary event to the one in the above equation ($S \geq \frac{\Delta(\epsilon, X)}{4}|y_2|$), and plugging in the definition for $S$, we get:

$$P\left(\Sigma_{i=1}^{|y_2|}\log\left(P_0^i + \left(1 - P_0^i\right)e^{X_i}\right) > \frac{1}{4}|y_2|\cdot\Delta(\epsilon, X)\right) \geq 1 - \exp\left(-\frac{|y_2|\sigma^2}{M^2}h\left(\frac{3\Delta M}{4\sigma^2}\right)\right) \tag{46}$$

Let $\delta > 0$, then for:

$$|y_2| > \frac{M^2}{\sigma^2 h\left(\frac{3\Delta M}{4\sigma^2}\right)}\cdot\ln\frac{1}{\delta} \tag{47}$$

We obtain from equation 46 with probability of at least $1 - \delta$ that $\Sigma_{i=1}^{|y_2|} \log \left( P_0^i + \left( 1 - P_0^i \right) e^{X_i} \right) > \frac{1}{4} |y_2| \cdot \Delta(\epsilon, X)$. Plugging this back into equation 36:

$$P(y_2 | x \oplus y_1) < P(y_2 | x_2) e^{-\frac{\Delta}{4} |y_2|} \tag{48}$$

With probability $1 - \delta$.

Using the same idea for $y_1$, we obtain:

$$P(y_1 | x) < P(y_1 | x_1) e^{-\frac{\Delta}{4} |y_1|} \tag{49}$$

Thus together, if $|y_1| + |y_2| > \frac{M^2}{\sigma^2 h\left(\frac{3\Delta M}{4\sigma^2}\right)} \cdot \ln \frac{1}{\delta}$, we have with probability $1 - \delta$ that:

$$P(y_1 \oplus y_2 | x) < P(y_1 | x_1) P(y_2 | x_2) e^{-\frac{\Delta}{4}(|y_1| + |y_2|)} \tag{50}$$

## B    PROOF OF THEOREM 1

We state theorem 1 with the full details:

**Theorem 2** *Let $\epsilon, \delta \in (0, 1)$, and $N, M > 0$. Let $x$ be a compositional problem, with $x_1$, $x_2$ being the corresponding sub-problems. Denote by $L_1, L_2$ the minimal solution length to $x_1, x_2$ respectively, and the total number of solutions to $x$ by $N$. Define $\Delta, \sigma$, the renormalizing term's mean and variance (as defined in equations 6 and 8 respectively) and by $M$ the bound on the logit noise (assumption 1). Under the assumptions of lemma 1, they are strictly positive, $\Delta, \sigma, M > 0$, and if the minimal solution length $L_1 + L_2$, satisfies $L_1 + L_2 > \frac{M^2}{\sigma^2 \cdot h\left(\frac{3\Delta \cdot M}{4\sigma^2}\right)} \ln \frac{N}{\delta}$, where $h(x) = (x + 1) \ln(1 + x) - x > 0$, we have with probability of at least $1 - \delta$ that the generation complexity (definition 2) satisfies:*

$$N(P, x) \geq N(P, x_1) N(P, x_2) \cdot e^{\frac{\Delta \cdot (L_1 + L_2)}{4}} \tag{51}$$

*Proof:*

Now, suppose there are $N$ solutions to the problem, all of length $\geq L$ (larger than the minimal description length), then we need to use a union bound. We get the result of the lemma 1 over all sequences with probability:

$$\left( 1 - \exp\left( -\frac{L\sigma^2}{M^2} h\left( \frac{3\Delta M}{4\sigma^2} \right) \right) \right)^N \geq 1 - N \exp\left( -\frac{L\sigma^2}{M^2} h\left( \frac{3\Delta M}{4\sigma^2} \right) \right) \tag{52}$$

Require this to equal:

$$= 1 - \delta \tag{53}$$

Thus for:

$$|y_1| + |y_2| > \frac{M^2}{\sigma^2 h\left(\frac{3\Delta M}{4\sigma^2}\right)} \cdot \ln \frac{N}{\delta} \tag{54}$$

We obtain with probability of at least $1 - \delta$ that all solutions satisfy the result of lemma 1. If the minimal solution length is $L_1 + L_2$, for all solutions to satisfy the inequality, we require:

$$L_1 + L_2 > \frac{M^2}{\sigma^2 h\left(\frac{3\Delta M}{4\sigma^2}\right)} \cdot \ln \frac{N}{\delta} \tag{55}$$

Thus we have with probability $1 - \delta$ that:

$$N(P, x) = \frac{1}{\sum_{y_1, y_2 \in correct\ solutions} P(y_1 \oplus y_2 | x)} \geq \tag{56}$$

$$\geq \frac{1}{\sum_{y_1, y_2 \in correct\ solutions} P(y_1 | x_1) P(y_2 | x_2)} e^{\frac{\Delta}{4}(L_1 + L_2)} = \tag{57}$$

$$= \frac{1}{\sum_{y_1 \in correct\ solutions\ for\ x_1} P(y_1 | x_1) \sum_{y_1 \in correct\ solutions\ for\ x_2} P(y_2 | x_2)} e^{\frac{\Delta}{4}(L_1 + L_2)} \tag{58}$$

$$= N(P, x_1) N(P, x_2) e^{\frac{\Delta}{4}(L_1 + L_2)} \tag{59}$$

## C  PROOF OF VARIANCE BOUND

Here we show that the variance of the noise $\sigma^2(p, X)$ is maximal for $p = \epsilon$.

**Lemma 4** *For $p \in (\epsilon, 1 - \epsilon)$ and a random variable $X$, it holds that:*

$$Var_X[\log(p + (1 - p)e^X)] \leq Var_X[\log(\epsilon + (1 - \epsilon)e^X)] \tag{60}$$

*proof:*

Consider the transformation $T_p(x) = \log(p + (1 - p)e^x)$, notice that its derivative *w.r.t.x* is:

$$\frac{dT_p}{dx} = \frac{(1 - p)e^x}{p + (1 - p)e^x} \tag{61}$$

Which always takes values in $(0, 1)$. Thus due to strict monotonicity, it is an invertible map for any $p \in (\epsilon, 1 - \epsilon)$. Additionally, for any $x$, we have:

$$\frac{dT_p}{dx} = \frac{(1 - p)e^x}{p + (1 - p)e^x} \leq \frac{(1 - \epsilon)e^x}{\epsilon + (1 - \epsilon)e^x} = \frac{dT_\epsilon}{dx} \tag{62}$$

Next, we look at:

$$Var_X[T_p(x)] = Var_X[T_p(T_\epsilon^{(-1)}T_\epsilon)(x)] = Var_X[(T_p \circ T_\epsilon^{(-1)})(T_\epsilon(x))] \tag{63}$$

Now, for an $M$-Lipschitz map, $T$, we have $Var_X[T(X)] \leq M^2 Var[X]$, therefore:

$$Var_X[T_p(x)] \leq \sup_x |\frac{d(T_p \circ T_\epsilon^{(-1)})}{dx}|^2 Var_X[T_\epsilon(x)] \tag{64}$$

Since the map $T_p \circ T_\epsilon^{(-1)}$, has a derivative that is bounded by:

$$|\frac{d(T_p \circ T_\epsilon^{(-1)})}{dx}| \leq \frac{|\frac{d(T_p)}{dx}|}{|\frac{d(T_\epsilon)}{dx}|} \leq 1 \tag{65}$$

We obtain:

$$Var_X[T_p(x)] \leq Var_X[T_\epsilon(x)] \tag{66}$$

Plugging in the definition of $T_p$, we get:

$$Var_X[\log(p + (1 - p)e^X)] \leq Var_X[\log(\epsilon + (1 - \epsilon)e^X)] \tag{67}$$

As desired.

## D  PROBABILITY BOUND FOR NOISY DECODING

In the presence of logit noise, the probability of each token in a decoding step is changed to:

$$P(i|context) \rightarrow P'(i|context) \leq \frac{P(i|context)}{P(i|context) + (1 - P(i|context))e^X} \tag{68}$$

Where $X$ is defined as in equation 4. The proof follows the proof of lemma 1 from the text before equation 32 up to equation 35. The main idea is to write the probability as a softmax over the noisy logits, then extract the original logits from the noise, bound the change in logits due to the noise using Jensen's inequality, and obtaining a bound in terms of the probability without noise.

The advantage of using this form of bound, is that it is relatively succinct, as it removes the dependence on the exact projection of noise on each token in the vocabulary, and instead takes into account the average noise on the different tokens. This allows to efficiently bound the change in probabilities of full sequences due to the noise.

# E EXPERIMENTAL DETAILS

**Composite Problem Construction:** The composite problems were created by pairs of problems in the formats described in subsection 5.1. Our results were based on 50 such composite problems for each experiment. The non-composite problems from human eval and code contests were also tested independently, and we used problems with standalone pass rates $\geq 0.1$, in order to avoid sampling too many solutions in the composite problems (which would typically have accuracy smaller than the product of pass rates of the standalone problems).

**Code Generation:** For each problem, code was generated by sampling at $T = 1$, and nucleus sampling with $p = 0.95$.

**Evaluation of Generated Code:** To evaluate the correctness of code generated by the LLM in the experiment described in subsection 5.1, we tested the code on the test cases provided in the datasets.

**Format of synthetic solutions to composite problems:** In subsections 5.2 (exponential length dependence) and 5.3 (assumptions), we performed a forward pass of the problem+solution in a scenario with and without composition in order to compare the logits in the two cases. To create correct solutions to the problems, that are "neutral" (not more likely to be generated by the LLM in a compositional problem than in the standalone case, or vice versa), we created solutions to the composite problems from the standalone problem solutions provided in the datasets. Typically, the model attempted to solve the problems sequentially, either by building functions to solve each problem in the pair, and apply the function sequentially, or by writing the explicit solutions one after the other. We used both templates to create solutions for these experiments.

**Sequence Probability Calculation:** In subsection 5.2, we measured the probability of solutions to problems with composition vs without composition. As explained above, we used to templates for the calculation that are similar to the model's generations (solution in a functional form, where a function is defined for each problem, and in a non-functional form, where the solutions to the problems are written sequentially). In both templates we observed similar trends. In order to avoid an artificial difference between composition and non-composition in the sequence probabilities due to the templates, we measured the probability of sequences after the first few tokens, so that the model has "time" to adjust to the format in both cases (composition and non-composition), and only measure the probabilities of the actual solution. We did this for both solutions of the compositional problem, for a more fair comparison between composition and non-composition sequence probabilities.

**Logit Noise Experiment:** In subsection 5.3, we used the templates as mentioned in the above on the sequence probability calculation. We extracted the of the logits of the solutions both in the case of composition and non-composition, and calculated the logit noise as defined in subsection 5.3 and equation 4.

**Additional Human Eval Template:** In addition to the template presented in the main text, we also tested the following template for Human Eval – Problem 1 has a True/False output, Problem 2 has an arbitrary output. Composition is to solve problem 1, then if its output is true, print the second problems' output, otherwise, print "-1":

> Complete the two following functions in python. Function 1: [Function 1 description]. Function 2: [Function 2 description]. If the output of the first function is "True" print the output to the second function. Otherwise print "-1".

# F    EXPERIMENT ON LLAMA-3-70B-INSTRUCT

In order to test the dependence of compositional hardness on model size, we repeat the experiment comparing generation complexity with vs without composition (described in subsection 5.1). As seen in figure 4, in most cases, composition requires more generations relative to the non-compositional case, but the ratio is typically smaller than before.

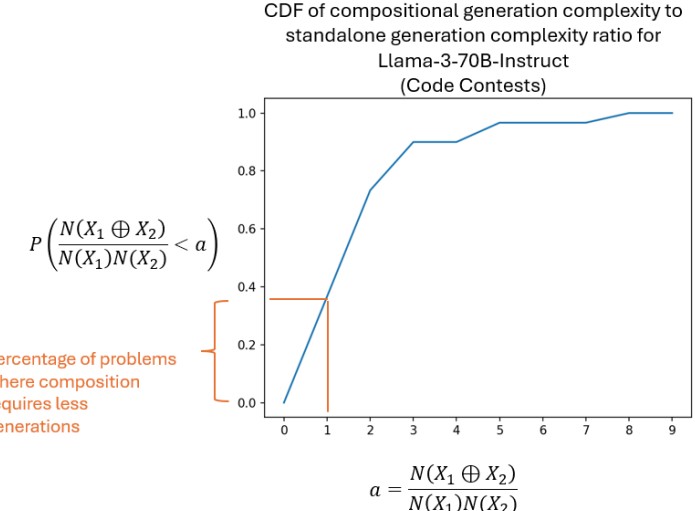

$$a = \frac{N(X_1 \oplus X_2)}{N(X_1)N(X_2)}$$

Figure 4: Cumulative distribution function for the ratio of generation complexity using composition, $N(P, x)$, to product of generation complexities for the standalone problems, $N(P, x_1) \cdot N(P, x_2)$ (corresponding to the multi-agent generation complexity). The x axis denotes values for the ratio of generation numbers required to solve the problem in the two cases (composition vs multi-agent), the y axis is the percentage of problems in which the ratio is no larger than this value (*e.g.* for $a = 5$, the y axis value is the percentage of problems where composition requires up to $\times 5$ more samples than the multi-agent case). As can be seen in most of the cases, composition requires twice more samples, and for some problems 10 times more samples.

However, if we increase the difficulty by concatenating four problems in the context and ask the model to only solve two of them (which keeps the compositional problem effectively a concatenation of two problems):

> Consider the following problems in python. Problem 1: [Problem 1]. Problem 2: [Problem 2]. Problem 3: [Problem 3]. Problem 4: [Problem 4]. Write a program that reads the inputs to the third and second problems sequentially and prints the outputs sequentially.

We once again obtain large ratios of compositional generation complexity, as seen in figure 5. This demonstrates the model's difficulty in extracting only the relevant information within the context for solving the problems (even when the separation is explicit), and that the noisy context increases the composition difficulty, such that even a concatenation of two problems becomes significantly more difficult. This is in accordance with our theory, where we consider a problem $x$, which is implicitly decomposable into two problems $x_1, x_2$, whose concatenated solutions, $y_1 \oplus y_2$, solves the problem. We obtain that the generation complexity to $x$, is much higher than the product of generation complexities for $x_1$ and $x_2$ due to the noise from the context.

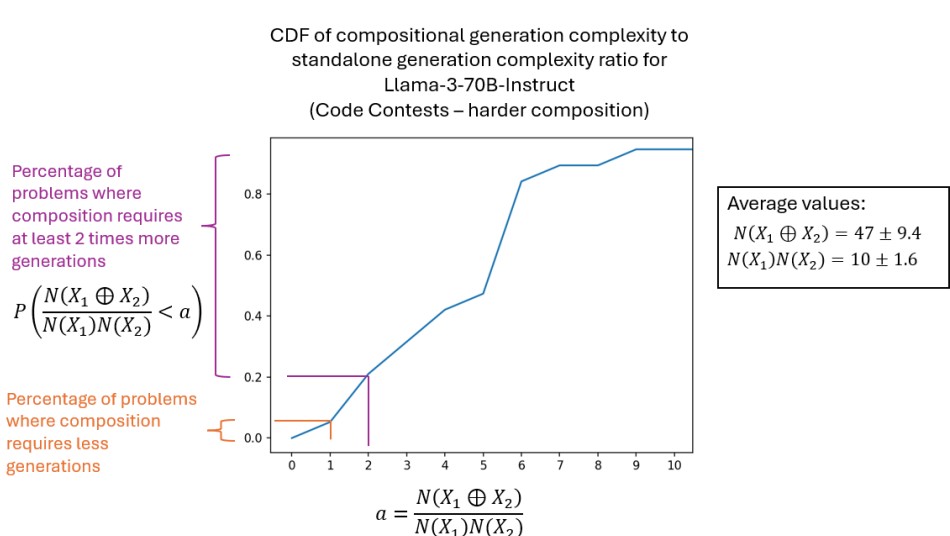

Figure 5: Cumulative distribution function for the ratio of generation complexity using composition, $N(P, x)$, to product of generation complexities for the standalone problems, $N(P, x_1) \cdot N(P, x_2)$ (corresponding to the multi-agent generation complexity). Here we explicitly ask the model to concatenate the solution to two problems, but also expose it to two additional ones that it is not meant to solve. The x axis denotes values for the ratio of generation numbers required to solve the problem in the two cases (composition vs multi-agent), the y axis is the percentage of problems in which the ratio is no larger than this value. As can be seen in most of the cases, composition requires at least 5 times more samples.

# G    NUMERICAL ESTIMATION OF $\Delta$ AND $\sigma$

To estimate $\Delta(\epsilon, X)$ we approximate the noise as a Gaussian with mean 0 and try two values standard deviation $\sigma = 1$ and $\sigma = 2$, then calculate $\Delta(\epsilon, X) \approx mean_{\{X_i\}}[\epsilon + (1 - \epsilon)e^X]$. The values are presented in figure 6, and match the estimation of $\Delta$ in the range of 0.05 to 0.2 from the above subsection. Similarly, we estimate $\sigma(\epsilon, X)$. The values are plotted in figure 6, showing typical values of $\sigma \approx 1 - 2$

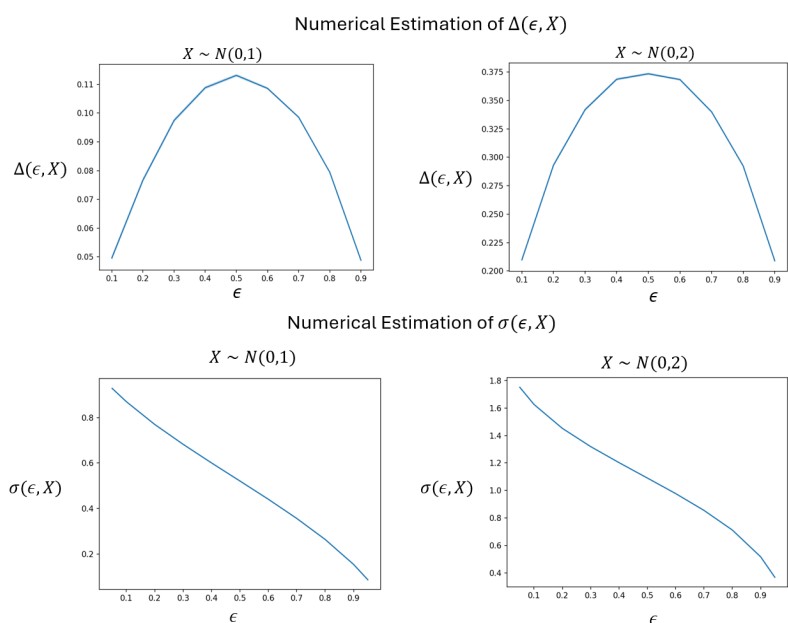

Figure 6: Numerical Estimation of $\Delta(\epsilon, X)$ and $\sigma(\epsilon, X)$.

