# OpenReview forum: "Compositional Hardness of Code in Large Language Models - A Probabilistic Perspective"
_ICLR.cc/2025/Conference — Submitted to ICLR 2025_

### Official Review · Reviewer_4RcV · 2024-10-17

**Soundness:** 3
**Presentation:** 3
**Contribution:** 3
**Rating:** 6
**Confidence:** 2

**Summary:**

This paper develops a complexity metric to measure the composition hardness of complex code generation by LLMs. By this metric, the authors conclude that decomposed individual generation by multiple agents is easier than generation within one context. This claim is further supported by empirical evidence.

**Strengths:**

originality: The paper provides a novel complexity metric that quantifies code generation difficulty. This is very interesting as the hardness of natural language processing tasks (code generation being one of them as part of instructions/generations can be in natural language) tends to be quite difficult to quantify.

quality: the paper provides both theoretical and empirical evidence for why multi-agent systems should be favored against single-agent generation when it comes to composed code generation problems. This is sound and clear.

clarity: The paper has a clear argument and consistent storyline.

significance: This paper is theoretically and empirically interesting in that it provides a theory to quantify the code generation hardness and also provides experiments to prove it.

**Weaknesses:**

The math part is too compact and creates difficulty for readers. The authors might consider providing more straightforward intuition and re-organizing some details in the appendix.

**Questions:**

1. Why is there no related work section?

2. It is not unknown that LLMs insufficiently utilize some part of the long context (e.g., [1]). How do you think this incapability to handle information in a long context window confounds your observation or LLM's incapability in doing complex code generation in one turn?

[1] Liu, N. F., Lin, K., Hewitt, J., Paranjape, A., Bevilacqua, M., Petroni, F., & Liang, P. (2023). Lost in the Middle: How Language Models Use Long Contexts. CoRR abs/2307.03172 (2023). arXiv preprint arXiv:2307.03172, 10.

---

> ### Author Response · Authors · 2024-11-21
>
> Thank you for your helpful feedback.
>
> Reply to weaknesses:
> - Thank you for your suggestion, we moved technical details from lemma 1 and theorem 1 to the appendix to make them more accessible in the revised version.
>
> Reply to questions:
> - The related works were discussed in the introduction, but to crystalize our contributions on top of existing works, we added a related works section.
> - We refer to works that show utilization of long context is limited (e.g. [1]), we will also cite the work you suggest. However, there is a big difference between our work and the works mentioned, because the contexts in our work are not necessarily very long, (can be less than 1000 tokens and not multiple documents). This is not considered a long context for most models. This exemplifies that the LLM does not just suffer from utilizing information in long contexts but also has trouble when dealing with multiple tasks within one context, getting noise from one task while trying to solve another. We will emphasize this difference to the existing works mentioned in the related works.
>
> [1] https://arxiv.org/abs/2404.06654
>
> [2] https://arxiv.org/abs/2307.03172

---

### Official Review · Reviewer_ka9k · 2024-10-18

**Soundness:** 2
**Presentation:** 3
**Contribution:** 3
**Rating:** 6
**Confidence:** 3

**Summary:**

The paper explains how if an LM is solving 2 coding problems (x1 and x2) at the same time, and since the generation of a solutions is probabilistic based on the temperature, then assuming the probability of finding a correct solution for solving each x1,x2 separately is P1,P2 (i.e. prob of the LLM solving x1 alone in 1 roll-out is P1). Then to solve x1 and x2 coding problems at the same time, at best you’d expect that getting both correct solutions in 1 roll-out is P1*P2. But since the LLM has finite capacity of depth, number of heads, and imperfect attention, then the authors find that the LLM does far worse than P1*P2, because when solving x1 and x2, the solution written for x1 gets noise from seeing the x2, and vice versa.

The paper advocates using an agentic approach and calling the LLM separately for each problem x1 & x2, which allows the LLM to use all its capacity and attention more optimally to solve the problems, so the success rate is P1*P2, and not lots worse.

The paper makes some assumptions and calculates some bounds on how much worse the accuracy of solving x1 & x2 together can be.

The paper shows results on Llama3-8b-Instruct

**Strengths:**

I like the presentation and analysis done on the problem with COT causing each step to be less likely to succeed if not done in an agentic way, intuitively most folks know completing multi-step tasks successfully has an exponentially small chance of success if all steps must succeed. But this work emphasizes that the prob of success is even worse than that – the multiple steps compete for attention and processing capacity in the layers – reducing chance of success even more – as the chance of each step succeeding is reduced by seeing all the other steps in the context.

I thought the paper was well written, and had lots of references to interesting relevant papers

I agree with the premise of the paper overall, the agent approach does seem intuitively better to eliminate distractions of irrelevant tokens  and empirically that is shown in the experiments in the paper.  I wish there was more empirical data for larger models showing how larger models maybe handle more problems in some scaling law sort of way.

**Weaknesses:**

W1. I agree from your results that current Llama-8b-Instruct as trained today doesn't do this multiple tasks as well as doing each task in isolation, but maybe if the LLM were fine-tuned on a dataset of compositional tasks they would learn to do screening/masking of irrelevant tokens more aggressively and not see the same degradation as currently seen, which might be because they are not typically trained on compositional tasks as often so haven't learned to aggressively mask/screen irrelevant tokens as well. Kind of impossible to test if training on more compositional data helps, so maybe not a fair weakness comment.

W2. I also wonder if the degraded performance is a model depth/size thing too – where bigger deeper models with more heads can mask/screen irrelevant tokens better and don’t see the degradation overload with multiple problems. It would be a stronger paper to understand either how bigger models do with multiple combined problems compared to solving them individually, and how much better models can do if finetuned on the compositional task so they might learn to mask/screen out irrelevant tokens better.

W3. Theoretically it's not clear to me the work rigorously proves the agent approach should be better for “all” datasets given the x1,y1 might be a few-shot example for the x2,y2 and help solve x2,y2 more accurately. I think the paper shows it doesn’t always help, wasn’t sure if that is just noise or are there reasons like x1 and x2 are so similar to help contextually.

W4. The assumptions about noise in the output layer, I’m unclear if for bigger deeper models if that still holds or if it the irrelevant tokens are masked out effectively on bigger models, or even the 8b models finetuned to do compositional tasks. The bounds analysis makes lots of assumptions that I'm not sure are valid in all cases.

**Questions:**

I look forward to reading other reviewers feedback and the authors responses and am open to adjust my scores based on more info.

Line 0: Have other papers done similar analysis say for multi-step tasks in general and this is a special case of that?

Line 147: If the LLM is effectively using attention to mask out the irrelevant parts in the lower layers, the higher layers could have noise=0.  So could the noise at L could be 0 in theory for a properly trained LLM for compositional problems?

Line 161: If x1 and x2 are similar – like “x1 = write a function that takes a number and returns the number + 7, x2 = write a function that takes a number and returns the number + 9” – then couldn’t their be a few shot effect that improves the prob of correct y2 given x1, y1, or even correct y1 given very similar x1,x2.  Does the semantic difference/similarities of x1,x2 impact how much worse the prob of a correct answer to both together is than the product of getting a correct answer to each question in isolation?

Line 304: Llama-3-8B Instruct for the model – do the results hold for other models?  Does this work show any interesting scaling law effects for larger Llama models?  Or on the best OpenAI or Anthropic models?  On larger models I would expect composition wouldn’t be as big of an impact, there would be less noise and sort of predictably less noise as the model size scales up. I could imagine that for problems of a certain complexity there is a number of problems that can be solved together before they interfere and start significantly degrading combined performance compared to solving each alone, and that number that can be solved together is proportional to model size in some way.

---

> ### Author Response · Authors · 2024-11-21
> **Reply to weaknesses**
>
> Thank you for your insightful comments.
>
> Reply to weaknesses:
> - We agree that if a model is finetuned on composing specific types of problems, then it may improve its performance on composing similar problems. Though we can then redefine the composite problem to be the basic problem (as that can essentially be treated as one problem that the model trained on) and try to compose them. Thus our interest is in composition in the sense of out of distribution performance, i.e. – If a model knows how to solve two problems, what can we say about its ability to solve their composition if it was not explicitly trained to do so. The significance of this is that since LLMs can solve such a wide range of problems, it would be beneficial to leverage it to solve problems which require combining different pieces of knowledge. Our work shows a limitation of this ability to combine its capabilities in a single context, pointing out that to utilize the model's capabilities better, splitting to contexts or finding a way to mask irrelevant information is important.
> - That is a good point –  We performed the composition experiment for the llama-3-70B-instruct model on code contest problems to see the dependence on size. The results are presented in appendix F, where indeed we see it is significantly more efficient at composition, though the effect is not diminished, and is similar to llama-3-8B-instruct’s results on humaneval composition. However, while in the experiments the compositions were explicitly concatenations of the subproblems, which are the simplest compositions, the theoretical results apply also to cases where the composition may be less direct, where there will be a mixing of tokens from different tasks that can be more challenging to separate, leading to a higher generation complexity.
> - It is a good point that potentially if the model generates $y_1$, then it can serve as an in-context learning example that would increase the chance of producing $y_2$. We discuss this in the revised version:
>   1. In reality, compositions may not necessarily be similar enough logically to induce in context learning - for example, if a compositional problem has a dynamics programming part and a randomized algorithm part, then there is no reason for one to improve the success rate of the other.
>   2. Our theoretical results do not guarantee that for any dataset composition becomes harder, it requires the noise to be large enough and the solutions to be long enough. Indeed, in the simpler experimental dataset of combined human eval problems (figure 1a), there are problems where composition is easier (indicating in-context learning), while for the more complex problems such as in the dataset of combined code contest problems (figure 1b), this happens less. Thus, there may be a competing mechanism of in context learning, that is overcome by the noise when complexity rises. We also note that, there is a significance to the order in which $x_1,x_2,y_1,y_2$ appear. In the case of in-context learning, the model is given $x_1,y_1,x_2$ and the model only needs to generate $y_2$, while in our compositional case, $y_1$ and $y_2$ need to be written in one program, and are not completely separated blocks of code, which may make it harder for the model to infer that $x_1,y_1$ are an in-context example to $x_2$ (after the generation of $y_1$).
> - Generally, it is experimentally observed even on the largest of models that longer contexts reduce their accuracy on tasks such as retrieval [1,2]. Specifically, performance depends on relevant information being positioned at “convenient” places within the context. Thus in compositional problems, even the larger models may not attend to the correct parts of the context, leading to the noise described in our framework. As exemplified in our experiments on the 70B model, while the decrease in performance is not as great as the 8B model, there is a consistent drop in accuracy on most compositional problems.
>
> [1] https://arxiv.org/abs/2404.06654
>
> [2] https://arxiv.org/abs/2307.03172

---

> > ### Author Response · Authors · 2024-11-27
> > **Additional note on the applicability to larger models**
> >
> > Additional note regarding the experiments on the 70B model - In appendix F, we also show that while the original compositional problems show a weaker effect on the 70B model relative to the 8B model (though still existent), with slightly harder compositions, the 70B model also shows the same trend of large compositional generation complexity relative to the non-compositional case.
> > The harder compositions are still effectively a concatenation of two problems, but additional noise is inserted to the composite problem by appending to the context two extra problems that the model is not asked to solve.
> > Such that four problems are presented, and the model is instructed to concatenate two of them, meaning the composite problem, $x$, is effectively decomposable to two subproblems, $x_1$, $x_2$, and the concatenation of their solutions, $y_1\oplus y_2$, solves the problem, as in our theoretical framework.
> > This shows that even in a much larger model, context noise can make a composite problem have a much higher generation complexity than the product of generation complexities of the subproblems, as in our theoretical results.
> > Meaning that while larger models are expected to be better at a given composition than smaller models, they still suffer from this effect as the compositions get harder from a context processing perspective, as proposed in our theoretical framework.

---

> ### Author Response · Authors · 2024-11-21
> **Reply to questions**
>
> Reply to questions:
> - Line 0: As written in the introduction – some previous works studied the advantages and limitations of multi-step tasks in language models, while primarily focusing on expressibility or learnability of deterministic functions that perform well defined mathematical tasks, such as the bit parity problem [3], Turing machines [4,5], or iterative composition of a single function [6], which cannot necessarily be translated in an interesting manner to code generation. In such results, for a given problem, the model provides a deterministic answer to the problem which is either correct or incorrect. This deviates from the practical use of language models for code generation that is based on probabilistic sampling of multiple solutions and filtering the correct ones (which can be 1 in 1000 or even worse). Thus previous works cannot capture the usefulness of LLMs on code, as most SOTA LLMs do not deterministically provide correct solutions to complex coding problems, but rather what makes them useful is sampling many solutions so that one is good. For this reason, we introduce the generation complexity metric that allows to quantify this usefulness by not just providing a binary answer to whether a model will succeed in a composition or not, but quantify *how hard* it is relative to the subproblems with a softer definition, which cannot be done in the previous settings. To improve accessibility and make this point clear, we summarized this in a new “related works” section.
> - Line 147: It may be possible to achieve better results on composition with training. Though as mentioned above in reply to weaknesses, the main motivation for composition is an out of distribution result – so that the model can piece together different problems it was not trained to solve together. An interesting future direction to alleviate this problem is a more “focused” attention mechanism that removes irrelevant context more aggressively and improves the signal to noise ratio, such as in [8].
> - Line 161: As mentioned above in reply to weaknesses, a few-shot effect is expected to rise when the problems are semantically similar, while in reality interesting compositions can have quite different subtasks. But we agree that studying the tradeoff between in-context learning to compositional hardness is a very interesting subject that we leave for future work.
> - Line 304: As mentioned above, we performed the experiment on the larger 70B model, and show that the effect is weaker on the same dataset relative to the 8B model, but existent.
>
> [3] https://arxiv.org/abs/2204.02892
>
> [4] https://arxiv.org/abs/2309.06979
>
> [5] https://arxiv.org/abs/2310.07923
>
> [6] https://arxiv.org/abs/2402.08164
>
> [7] https://arxiv.org/abs/2410.05258
>
> [8] https://arxiv.org/abs/2410.05258

---

> ### Author Response · Authors · 2024-11-26
> **Invitation for comments**
>
> Dear reviewer ka9k,
>
> We greatly value your constructive feedback on our paper. We would greatly value your review of our responses to ensure we have properly addressed your concerns.

---

### Official Review · Reviewer_xEcp · 2024-10-30

**Soundness:** 2
**Presentation:** 1
**Contribution:** 2
**Rating:** 5
**Confidence:** 2

**Summary:**

The paper addresses a natural problem: how good are LLMs at solving programming tasks when those tasks require compositions of subsolutions? Given the rise of co-pilots, this is extremely relevant. The paper presents a simple mathematical model of the problem, and some experimental results.

While I think the problem is natural, I found the realisation in the present paper to be weak. The formal model is poorly presented (Definition 1 is effectively empty, the theoretical results are laboured and their significance very unclear). The experimental results section is the most interesting, but I found it somewhat contrived and unconvincing.

Overall, my sense is this is an interesting direction, but its a rather strange paper overall.

**Strengths:**

Very compelling and natural problem with realworld implications.

**Weaknesses:**

Theoretical results are poorly presented and laboured. Significance not clear.

Experimental results seem contrived and the experimental setup seems a little arbitrary. Significance of these results not clear.

**Questions:**

Can you explain the significance of your theoretical results in informal terms?

Can you better justify the experimental set-up and the decisions you made there?

Do you have a clear hypothesis to explain your experimental results?

---

> ### Author Response · Authors · 2024-11-21
>
> Thank you for your feedback.
>
> Reply to weaknesses:
> - We explain below the significance of the theoretical results.
> - Below we explain how the experimental setup arises from the theoretical results, why it is not arbitrary, and the significance of the results.
>
> Reply to questions:
> - Significance explained informally –
>   - Why generation complexity is important - As presented in the introduction, when it comes to LLMs generating code to solve problems, a common practice is to sample multiple programs (could be 100, could be 1000, or even more), and to filter out those that pass correctness tests. There are instances where out of thousands of generations only one solution is correct. Due to the use of testing units, an LLM can still be useful even if only 1 in 1,000 sampled programs is correct, because you can filter out the correct one. Of course, if 1 in 100 is correct, then the LLM is much more useful, as you would only need a tenth of the sampling budget/test time compute (which translates to a tenth of the actual price in standard LLM APIs). Understanding how many programs one needs to sample from an LLM to assure a correct solution is sampled, quantifies the LLM's usefulness on such a task. The generation complexity metric quantifies exactly this.
>   - The gist of our results - We then show that in tasks that involve several steps, one would need to sample exponentially more programs if it is done sequentially within one context, compared to sampling the solutions separately, in different contexts. This quantifies a difficulty of problem composition. It is significant for the real purpose of code generation, because if one can sample a correct solution with a budget of 100 generations compared to 1,000 generations, the cost is significantly lower.
>   - Where this comes into play in real life - When dealing with compositional coding problems, one typically uses a chain of thought to solve a series of subproblems whose concatenated solutions solve the problem y_1⊕…⊕y_n (e.g. break the task into multiple helper functions). It is not intuitive that after decomposing the problem, an LLM that can solve one subproblem with probability 0.5 (1 in 2 generations on average) and another with probability 0.2 (1 in 5 generations on average), will have success probability 0.01 (1 in 100 attempts) to solve them in the same context and not the product of individual probabilities 0.2*0.5=0.1 (1 in 10 generations). This phenomena of in-context hardness of composition is what we tackle in this paper.
>   - We added a related works section that highlights the motivations for our theoretical approach given existing works.
> - We do not claim that this covers the entire aspect of composing problems, but that the aspect we present can be a very significant contributor to LLM failure in composition, which has not been covered by previous works, and that a multi-agent system can have a big advantage in this regard.
> - Experimental justification – as the aspect of compositional hardness we want to tackle is the in-context hardness explained above, the experimental setup is natural, as you take two problems, and observe that solving them together requires more generations than independently solving them in separate contexts. Thus concatenating problems can be thought of as a lower bound for the hardness of composition (as they are an easy form of composition) - meaning harder compositions will have higher complexity. We added this point to the revised paper.
> - Hypothesis to explain experimental results – as mentioned in the paper, when combining two problems, the self-attention mechanism mixes attention in tokens from different problems. This is quantified by the noise we defined – if the hidden representations in the non-compositional case are more "accurate" in the sense that they lead to sampling more correct programs, then in the compositional case, the representations are noised, which leads to the generation of less accurate programs.

---

> ### Author Response · Authors · 2024-11-26
> **Invitation for comments**
>
> Dear reviewer xEcp,
>
> We appreciate your feedback on our paper. To ensure our responses are helpful to resolve your concerns, we would greatly appreciate it if you could review them.

---

### Official Review · Reviewer_Q7TG · 2024-11-04

**Soundness:** 2
**Presentation:** 3
**Contribution:** 3
**Rating:** 6
**Confidence:** 4

**Summary:**

This submission presents a probabilistic model of compositional problem solving by LLMs (eq (2-3)). Based on this model, the authors present a theoretical result showing that solving the composition (for various notions of "composition") of two problems can be exponentially more complex (for a probabilistic notion of complexity related to the expected number of samples to reach success) than solving the two problems separately. The authors then present some empirical evidence justiying the assumptions behind their model as well as their main theoretical result.

**Strengths:**

The authors tackle a problem which is of high importance from the perspective of implementation and of explainable AI. Problem decomposition can be both an efficient method (as evidenced in this paper), and one that is easier to interpret (since part of the explaining is being done by the decomposition).

The methodology is elegant, providing what feels like the right level of abstraction to be able to make meaningful and general statements about systems as complex as LLMs. The formal statements are plausible (although there are some issues in the proofs, see below). The empirical evidence, particularly Fig 1 and Fig 2 is convincing (if rather limited).

**Weaknesses:**

* Conceptually, a more detailed examination of the notion of composition would probably be beneficial. Some "compositions" are clearly harder than others. Concatenating a function with itself is trivial, composing a function with itself typically isn't. Therefore, the "noise" term which forms the (very elegant) technical heart of the paper cannot be expected to be as agnostic to the composition mechanism as suggested in this work. This aspect of problem composition is not addressed at all.

* Technically, I'm not 100% convinced by the proof of Lemma 1. At Eq (40), Bennet's inequality is used. It says that \\(P(\sum_{i=1}^n X_i>t)\leq \exp\left(-\frac{n\sigma^2}{M^2}h\left(\frac{Mt}{n\sigma^2}\right)\right)\\). Here, the inequality is reversed and uses a term \\(1-\exp()\\). This is incompatible with the definition, and would be the lower-bound for \\(P(\sum_{i=1}^n X_i\leq t)\\). Also, when pattern-matching this definition with Eq (40), I could not find a reason/meaning for the term \\(L\\) and for the factor \\(3\\). So Eq (40) looks suspicious. The statement before Eq (42) is also suspicious (and I'm not talking about the typo). What is the probability of \\(1-\delta\\) associated with? Is it the probability of sampling \\( |y_2| \\) noise terms such that (42) holds? Can you explain how you go from Eq (35) to Eq (42)?

* Section 4.3 does not really serve any purpose, the noise model is completely ad-hoc (centered Gaussian with standard deviation \\(\sigma=1,2\\) and so it doesn't tell us anything.

* Specific comments:

- Def 1: Is \\(f\\) written in \\(L\\)? What exactly is the meaning of \\(y\\) implementing \\(f\\)?
- Def 2: What is \\(\Sigma\\)? Just below, "sampled from the distribution" -> the conditional distribution \\(P(\cdot\mid x)\\)?
- Eq (4) and throughout the document: I think you're writing the quotient symbol / when you mean the set difference \\(\setminus \\) symbol.
- Eq (13)-(14): use \\(\exp\\) instead of \\(exp\\)
- l292: \\(\delta \exp(8)\approx 3000\delta\\)
- l485: \\(\Delta \approx 0.05 - 0.2\\) looks like a subtraction, also it seems to contradict the earlier point that \\(\Delta\approx 0.2\\).
- Eq (22): this step is not necessary
- Eq (30): What is \\(V\\)? Where does it come from?
- Eq (31): I think all instances of \\(x\oplus y_1\oplus y_2\\) here should be \\(x_2\oplus y_2\\)
- Eq (33): I would find this step much clearer if you invoked Jensen's Inequality which does the job exactly.
- Eq (34): Minus sign missing before \\(\log\\)

**Questions:**

1. Where does the functional shape of eq (5) come from? Why this shape and not another?
2. Can you answer all the questions above about the proof of Lemma 1?
3. In Fig 1 and Fig 2, exactly which of the two Human Eval composition is being used? What happens if you take consider the other?
4. What can you say about the relative compexity of different combination mechanisms and its impact on the noise term of Eq (2-3)?

---

> ### Author Response · Authors · 2024-11-21
> **Reply to weaknesses**
>
> Thank you for your detailed feedback.
>
> Reply to weaknesses:
> - Regarding the technical proof of the lemma. We are sorry for the confusion, we elaborated more on these steps in the proof in the revised version to improve clarity. For completeness we explain here briefly on the points that you raised:
>   1. Equation 40 (now numbered 46): Bennet's inequality bounds the deviation from the expected value, we use the lower tail bound $P(\sum_i (X_i-E[X_i ])≤-t)≤exp⁡()$ (symmetric to the upper tail bound case with ≥t). Move the expectation value to the r.h.s. and take $t=3/4 \sum_i E[X_i]$, which gives $P(\sum_i X_i≤1/4 \sum_i E[X_i])≤exp⁡()$. Lastly, take the complementary event, which gives the desired result $P(\sum_i X_i>1/4 \sum_i E[X_i])≥1-exp⁡()$.
>   2. The choice of t which includes the 3/4, explains the factor of 3/4 on the right hand side of equation 40. The factor L is a typo, meant to be $|y_2 |$, the length of the solution. The origin of this is that in Bennet's inequality, you place there the variance of the sum, which in our case is bounded by $|y_2 | σ^2$. To make the exponent small and obtain a large probability 1-δ, we go to equation 41 (now 47), where we find how large $|y_2 |$ needs to be for this to hold.
>   3. Thus before equation 42 (now numbered 48), indeed as you thought, 1-δ is the probability that the noise sampled is large enough so that equation 42 holds.
>   4. The transition from 35 to 42 (now 36 to 48): in 35, we have $P_{composition} < P_{no composition}⋅e^{-S}$, and in the series of equations 36-41 (now 37-47), we introduce a concentration inequality on S, showing that with high probability $S>Δ/4|y_2 |$. So, with high probability, $P_{composition}<P_{\textit{no composition}}⋅e^{-S}<P_{\textit{no composition}}⋅e^{-Δ/4|y_2 |}$.
>   5. See revised version for full details.
> - The subsection 4.3 serves two parts – first to plot the noise in a compositional coding problem and see that it qualitatively looks as we expect in the assumption (symmetric, continuous) and to estimate its magnitude. The second part, is to take noise with this typical size and shape, to get a ballpark estimation to Δ and σ.
> - Def 1: f is an abstract function that solves x (e.g. if x is the problem of finding shortest paths between two vertices in a graph, then f is a mathematical function that receives a graph and two vertices, and outputs the shortest path in the graph between them). The solution y, is the implementation of the abstract function f in the language L, i.e. the program.
> - Def 2: We are sorry for not clarifying, Σ is the vocabulary of the LLM, thus Σ* are sequences of tokens. "Sampled from the distribution" indeed refers to the conditional distribution P(⋅|x), as in definition 2 we sum over conditional probabilities of correct solutions P(y|x). For consistency we changed the notation of vocabulary from Σ to V in the revised version.
> - Set difference vs quotient difference – you are correct, we mean the set difference, we fixed it.
> - We changed the exp font.
> - The line "Δ≈0.05-0.2" means Δ is in the range 0.05 to 0.2, not a subtraction. We corrected this in the text. Hence it does not contradict the earlier point of Δ≈0.2, but agrees with it.
> - Eq 30: We referred earlier to V as the vocabulary, we clarified in the revised version.
> - Eq 31: Yes, you are correct, we fixed this.
> - Eq 33: Noted, we will use Jensen's inequality
> - Eq 34: Thank you, it is true, we corrected. We note that in the following we did use the correct sign.

---

> ### Author Response · Authors · 2024-11-21
> **Reply to questions**
>
> Reply to questions:
>
> 1. The functional shape of equation 5 comes from the reweighting of the logits due to the noise and applying Jensen's/AMGM inequality, as seen in the proof of lemma 1 up to equation 35. We added a derivation in appendix D to explain this more clearly. This represents a useful upper bound on the probability of producing a correct token in a compositional problem relative to the non compositional case. There can be other forms of bounds, but this is a relatively succinct bound as it removes the dependence on the exact projection of noise on each token in the vocabulary, and instead takes into account the average noise on the different tokens.
> 2. We hope the explanations above and clarifications in the revised version answer the questions on the proof of lemma 1.
> 3. In figure 1 and 2 we used the second form (multiplication). Both forms yielded similar results.
> 4. Relative complexity of different compositions - in the theoretical results, a solution to a compositional problem is the concatenation of solutions to the subproblems, which is consistent with the use of a chain of thought, in which a problem x is solved in steps $y_1⊕…⊕y_n$ (such as helper functions in programming). In our framework, the relative complexity of compositions is quantified by the length of the solution and the magnitude and shape of the logit noise introduced by the context (both in the prompt and partial generations of the model). This quantifies a specific type of compositional hardness introduced by context noise. Indeed, there may be other types of composition difficulties not captured by tuning the noise from our framework, but the aspect of hardness in this work serves as a lower bound on compositional hardness due to its presence in problems that require sufficiently long solutions. The experiments section shows this effect in concatenations, which are the simplest compositions, in order to cleanly show that the effect of the context noise is significant, hence they can be seen as a lower bound. We note that the results of our paper do not claim to capture all aspects of compositional complexity, but rather an unexpected mechanism introduced by the noise, which can be a significant factor of composition, due to the necessity to break down complex problems to smaller problems with chain of thought.

---

> > ### Author Response · Authors · 2024-11-26
> > **Invitation for comments**
> >
> > Dear reviewer Q7TG,
> >
> > We value your feedback on our theoretical work and updated the proofs according to your comments. To ensure we have properly addressed your concerns, we would greatly appreciate it if you could review our responses.

---

### Meta-Review · Area_Chair_gVqQ · 2024-12-19

**Metareview:**

This paper investigates the degradation in performance of language models when they are tasked with solving two programming problems jointly. They present a mathematical analysis of this degradation, and validate that it appears to be true for competition programming tasks.

A strength is that it tries to give a first principles analysis of the phenomena its studies. A weakness is that the papers conclusion, at least in qualitative form, is already known (see "Faith and Fate: Limits of Compositionality", among many other papers).

I argue for rejection because the basic insight is already well known, hence the lukewarm reception from the reviewers. I don't think that the theoretical analysis compensates for this, particularly because it is only applicable to a very specific kind of compositionality, namely when the function outputs are concatenated.

**Additional Comments On Reviewer Discussion:**

Missing related work was actually a big problem in the original submission, but this was addressed in the rebuttal. However this is not something that can be fully resolved through revision because the related work impinges upon the novelty of this paper.

---

### Decision · Program_Chairs · 2025-01-22

Reject